# Working memory gating in obesity is moderated by striatal dopaminergic gene variants

Nadine Herzog[1,2]\*, Hendrik Hartmann[1,3,4], Lieneke Katharina Janssen[1,5], Arsene Kanyamibwa[4], Maria Waltmann[1,6], Peter Kovacs[7], Lorenz Deserno[6,8], Sean Fallon[9], Arno Villringer[1], Annette Horstmann[1,3,4]

[1]Department of Neurology, Max Planck Institute for Human Cognitive & Brain Sciences, Leipzig, Germany; [2]International Max Planck Research School NeuroCom, Leipzig, Germany; [3]Collaborative Research Centre 1052, University of Leipzig, Leipzig, Germany; [4]Department of Psychology and Logopedics, Faculty of Medicine, University of Helsinki, Helsinki, Finland; [5]Institute of Psychology, Otto von Guericke University Magdeburg, Magdeburg, Germany; [6]Department of Child and Adolescent Psychiatry, University of Würzburg, Würzburg, Germany; [7]Medical Department III – Endocrinology, Nephrology, Rheumatology, University of Leipzig Medical Center, Leipzig, Germany; [8]Department of Psychiatry and Psychotherapy, Technische Universität Dresden, Dresden, Germany; [9]School of Psychology, University of Plymouth, Plymouth, United Kingdom

**\*For correspondence:**
herzog.nd@gmail.com

**Abstract** Everyday life requires an adaptive balance between distraction-resistant maintenance of information and the flexibility to update this information when needed. These opposing mechanisms are proposed to be balanced through a working memory gating mechanism. Prior research indicates that obesity may elevate the risk of working memory deficits, yet the underlying mechanisms remain elusive. Dopaminergic alterations have emerged as a potential mediator. However, current models suggest these alterations should only shift the balance in working memory tasks, not produce overall deficits. The empirical support for this notion is currently lacking, however. To address this gap, we pooled data from three studies (N = 320) where participants performed a working memory gating task. Higher BMI was associated with overall poorer working memory, irrespective of whether there was a need to maintain or update information. However, when participants, in addition to BMI level, were categorized based on certain putative dopamine-signaling characteristics (single-nucleotide polymorphisms [SNPs]; specifically, Taq1A and DARPP-32), distinct working memory gating effects emerged. These SNPs, primarily associated with striatal dopamine transmission, appear to be linked with differences in updating, specifically, among high-BMI individuals. Moreover, blood amino acid ratio, which indicates central dopamine synthesis capacity, combined with BMI shifted the balance between distractor-resistant maintenance and updating. These findings suggest that both dopamine-dependent and dopamine-independent cognitive effects exist in obesity. Understanding these effects is crucial if we aim to modify maladaptive cognitive profiles in individuals with obesity.

## eLife assessment

The present study provides **valuable** evidence on the neurochemical mechanisms underlying working memory in obesity. The authors' approach considering specific working memory operations (maintenance, updating) and putative dopaminergic genes is **solid**, though the inclusion of a more direct measure of dopamine signaling would have strengthened the work.

## Introduction

In order to function efficiently in a dynamic environment, we must be able to resist distractions while simultaneously being open to update information in response to evolving goals and task requirements. This tension demands a delicate balance, which is thought to be governed by one of our core cognitive control systems – our working memory (WM). Computational and neurophysiological theories propose a metaphorical 'gate' that regulates the access to WM (*Badre, 2012*; *O'Reilly and Frank, 2006*; *Chatham and Badre, 2015*). When the gate is closed, WM representations are isolated from perceptual input and interference is prevented. When the gate is opened, rapid updating is allowed. Evidence strongly implicates the prefrontal cortex (PFC) in distractor-resistant maintenance, while updating is thought to be supported by the striatum (*Miller and Cohen, 2001*; *D'Ardenne et al., 2012*; *Braver and Cohen, 2000*). Importantly, the neurotransmitter dopamine plays a crucial role in balancing these complementary processes. Within the PFC, tonic dopamine levels mediate maintenance in an inverted-U-shaped manner: very high and very low levels promote gate opening, while medium levels promote gate closing (*Durstewitz and Seamans, 2008*; *Cools and D'Esposito, 2011*). Within the striatum, phasic increases in dopamine are needed to signal WM updating (*Hazy et al., 2006*; *D'Ardenne et al., 2012*). Importantly, the effectiveness of phasic rises in dopamine to override PFC tonic signals depends on initial baseline dopamine levels in the striatum (*Cools and D'Esposito, 2011*; *van Schouwenburg et al., 2010*; *Ranganath and Jacob, 2016*). Decreases in tonic dopamine levels in the striatum seem to raise the threshold for updating signals, thus potentially hindering updating (*O'Reilly and Frank, 2006*). Supporting this, worse updating of WM contents can be observed in unmedicated patients with Parkinson's disease (*Fallon et al., 2017a*), older individuals (*Podell et al., 2012*), or more generally, in individuals with lower dopamine synthesis capacity (*Colzato et al., 2013*). Notably, dominance in one WM process typically comes at the cost of the other (*Dreisbach and Fröber, 2019*; *Fallon et al., 2017b*). Consequently, an individual's capacity to ignore or update (ir)relevant information may vary according to their baseline dopamine levels (e.g. *Cools and D'Esposito, 2011*; *Furman et al., 2021*; *Jongkees, 2020*).

Interestingly, the intricate relationship between dopamine levels and WM gating might be key in further understanding discrepancies in the literature regarding WM functioning in obesity. While many studies show reduced (general) WM in obese individuals (e.g. *Yang et al., 2020*; *Yang et al., 2019*; *Yang et al., 2018*; *Gonzales et al., 2010*; *Coppin et al., 2014*), there are others who do not find such associations (e.g. *Calvo et al., 2014*; *Schiff et al., 2016*; *Alarcón et al., 2016*). Based on the above considerations, these inconsistencies may be due to prior studies not clearly differentiating between distractor-resistant maintenance and updating in the context of WM. This distinction may be crucial, however, as indirect evidence hints at potential specific alterations in these two sub-processes in obesity. For instance, obesity has been associated with aberrant dopamine transmission, with there being an abundance of literature linking obesity to changes in D2 receptor availability in the striatum (see, e.g., *Horstmann et al., 2015*). However, results are not consensual, with studies reporting decreased, increased, or unchanged D2 receptor availability in obesity (*Ribeiro et al., 2023*; *Janssen and Horstmann, 2022*; see *Darcey et al., 2023*, for a potential explanation). Additionally, there are reports of differences in dopamine transporter availability in both obese humans (*Chen et al., 2008*; but also see *Pak et al., 2023*) and rodents (*Narayanaswami et al., 2013*; *Jones and Fordahl, 2021*; *Hamamah et al., 2023*). The observed changes in dopamine are often interpreted as being due to chronic dopaminergic overstimulation resulting from overeating (*Volkow and Wise, 2005*; *Volkow et al., 2008*) and altered reward sensitivity as a consequence thereof (*Blum et al., 1996*). Considering that WM gating is highly dependent on dopamine signaling, such changes could theoretically alter the balance between maintenance and updating processes in obesity. Next to this, obesity has frequently been associated with functional and structural changes in WM gating-related brain areas, implying another pathway through which WM gating might get affected. At the level of the PFC, studies have reported reduced gray matter volume and compromised white matter microstructure in individuals with obesity (*Debette et al., 2014*; *Kullmann et al., 2016*; *Morys et al., 2024*; *Lv et al., 2024*), and functional changes become evident with frequent reports of decreased activity in the dorsolateral PFC during tasks requiring cognitive control (e.g. *Morys et al., 2018*; *Xu et al., 2017*). Notably, *Han et al., 2022*, observed significantly lower spontaneous dlPFC activity during rest, potentially indicating reduced baseline dlPFC activity in obesity. On the level of the striatum, gray matter volume seems to correlate positively with measures of obesity (*Horstmann et al., 2011*), and individuals with obesity

show greater activation of the dorsal striatum in response to high-calorie food stimuli compared to normal-weight individuals, indicating a stronger dopamine-dependent reward response to food cues (*Stice et al., 2008*; *Small et al., 2003*). Additionally, changes in connectivity between and within the striatum and PFC in obesity, both structurally (*Li et al., 2023*) and functionally (*Verdejo-Román et al., 2017a*; *Contreras-Rodríguez et al., 2017*), have been reported. Although these studies mostly investigate brain function in relation to food and reward processing, changes in these areas may also impair the ability to adequately engage in WM gating processes, as activity in affective (reward) and cognitive fronto-striatal loops immensely overlap (*Janssen et al., 2019*). On the behavioral level, individuals with obesity consistently demonstrate impairments in food-specific (*Janssen et al., 2017*) but also non-food-specific goal-directed behavioral control (*Janssen et al., 2020*) and reinforcement learning (*Weydmann et al., 2024*). It seems that difficulties with integrating negative feedback may be central to these alterations (*Mathar et al., 2017*; *Käenmäki et al., 2010*), which could explain a potential insensitivity to the negative consequences associated with (over) eating. Crucially, in humans, a substantial contribution to (reward) learning is mediated by WM processes (*Moustafa et al., 2008*; *Collins and Frank, 2012*; *Collins and Frank, 2018*; *Collins et al., 2014*; *Collins et al., 2017*; *Westbrook et al., 2024*). The observed difficulties in reward learning in obesity may hence partly be rooted in a failure to update WM with new reward information, suggesting cognitive issues that extend beyond mere difficulties in valuation processes. However, empirical support for this interpretation is currently lacking. A more nuanced understanding of the effects of obesity on WM is crucial, however, as it could lead to more targeted intervention options.

In the present study, we therefore aim to examine potential obesity-dependent alterations in WM gating. To this end, we pooled together data on body mass index (BMI; kg/m²) and a WM gating task from three different studies conducted in our lab. In light of the behavioral and neuropharmacological findings discussed above, we hypothesized that individuals with a high BMI would display worse updating, potentially offset by enhanced distractor-resistant maintenance.

Given dopamine's central role in WM gating, such behavioral patterns might be driven by the altered dopamine signaling observed in obesity (as discussed above). However, in addition to this, inherent predispositions with respect to dopamine signaling may also contribute. In this context, several single-nucleotide polymorphisms (SNPs) related to dopamine transmission have garnered significant attention in recent years. For instance, catechol-*O*-methyltransferase (COMT) Val158Met activity primarily influences dopamine breakdown in the PFC (*Tunbridge et al., 2004*; *Sesack et al., 1998*), and carrying the Met allele of this SNP is associated with reduced COMT activity, leading to higher synaptic dopamine levels (*Bilder et al., 2004*). Consistent with this, individuals with the Met allele tend to perform better on tasks that require stable maintenance of WM representations compared to those with the Val allele (*Berryhill et al., 2013*; *Farrell et al., 2012*; *Savitz et al., 2006*). Furthermore, the Taq1A polymorphism has been associated with D2 receptor density in the striatum. A-allele carriers of this polymorphism exhibit lower receptor density and show distinct performance patterns on tasks involving WM updating (*Pohjalainen et al., 1998*; *Jönsson et al., 1999*; *Eisenstein et al., 2016*; *Stelzel et al., 2010*; *Persson et al., 2015*; *Li et al., 2019*). Interestingly, Taq1A and COMT have been demonstrated to interactively affect WM functioning (*Berryhill et al., 2013*; *Xu et al., 2007*; *Garcia-Garcia et al., 2011*; *Stelzel et al., 2009*; *Wishart et al., 2011*; *Persson and Stenfors, 2018*). Consequently, we aim to examine this interactive effect and assess if it varies with BMI. In addition, PPP1R1B and C957T gene polymorphisms have also been linked to WM (*Hotte et al., 2006*; *Ma et al., 2022*; *Smith et al., 2014*; *Xu et al., 2007*; *Klaus et al., 2019*; *Jacobsen et al., 2006*). The PPP1R1B polymorphism (rs907094) codes for dopamine and cAMP-regulated neuronal phosphoprotein (DARPP-32) – a protein that potently modulates dopamine D1-dependent synaptic plasticity in the striatum (*Ouimet et al., 1984*; *Calabresi et al., 2000*; *Lindskog et al., 2006*; *Girault and Nairn, 2021*). The C957T (rs6277) polymorphism, on the other hand, is known to impact dopamine D2 mRNA translation (*Duan et al., 2003*) and postsynaptic D2 receptor availability in the striatum (*Hirvonen et al., 2005*). Both polymorphisms have also been associated with (diet-induced) weight gain (*Sharma and Fulton, 2013*; *Hu et al., 2006*; *Müller et al., 2012*). Their effect on BMI-dependent WM gating, however, remains unknown. In order to test the impact of these four candidate polymorphisms, we also added participants' genetic information to our analyses. We hypothesized that BMI-dependent distractor-resistant maintenance and/or updating of WM representations would be modulated by (1) an interaction of COMT and Taq1A, a main effect of (2) DARPP-32, and/or a main effect of (3) C957T.

**Table 1.** Sample characteristics.

| Project | All | | | BEDOB | | | GREADT | | | WORMCRI | | |
|---|---|---|---|---|---|---|---|---|---|---|---|---|
| N (male) | 320 (166) | | | 156 (43) | | | 86 (86) | | | 78 (37) | | |
| | mean (sd) | min | max | mean (sd) | min | max | mean (sd) | min | max | mean (sd) | min | max |
| BMI | 26.38 (6.35) | 17.51 | 45.54 | 29.172 (7.695) | 17.51 | 45.54 | 24.025 (2.799) | 18.632 | 36.419 | 23.217 (2.735) | 18.929 | 29.888 |
| IQ | 105.41 (10.61) | 71 | 122 | 101.575 (11.979) | 71 | 122 | 109.151 (7.249) | 91 | 118 | 107.731 (10.416) | 74 | 118 |
| Age | 26.93 (6.82) | 12.17 | 49.75 | 26.879 (8.907) | 12.167 | 49.75 | 26.756 (4.474) | 18 | 40 | 26.799 (3.859) | 20.106 | 36.290 |
| DFS | 54.89 (11.61) | 33 | 97 | 55.839 (10.163) | 35 | 91 | 57.046 (15.107) | 33 | 97 | 50.584 (8.546) | 34 | 71 |

In addition to our primary investigations, we further conducted exploratory analyses on a subsample of our data. Two of the three studies had data available on the ratio of phenylalanine and tyrosine to other large neutral amino acids. This ratio represents the peripheral dopamine precursor availability and can be considered a potential proxy for central dopamine synthesis capacity (*Leyton et al., 2004*; *Montgomery et al., 2003*). Existing evidence suggests that this measure may be linked to WM performance in a diet-dependent manner (*Hartmann et al., 2020*). By looking at amino acid ratio and its connection to BMI-dependent WM gating, we sought to assess the possible influence of dopamine at the system level.

## Results
### Sample descriptives
Three participants were excluded from the analyses, as they performed below chance (<50% correct in all four conditions). One subject was excluded as they reported that they didn't perform the task properly during the post-task strategy assessment. The final sample thus consisted of 320 participants. The average age of the sample was 26.93 years (SD = 6.82, min = 12.17, max = 49.75). There were 166 males. Mean BMI was 26.38 kg/m² (SD = 6.35, min = 17.51, max = 45.54). Mean IQ was 105.41 (SD = 10.61, min = 71, max = 122). Data for Dietary Fat and free Sugar Questionnaire (DFSQ) was missing for five subjects. Mean DFSQ score was 54.89 (SD = 11.61, min = 33, max = 97). Please refer to *Table 1* for a list of full sample characteristics (per study).

### Taq1A genotype moderates the association between BMI and WM updating, independent of COMT
First, to test how BMI would relate to WM gating, we ran a logistic regression model predicting trial-based accuracy by the interaction of task condition and BMI. As expected, final results showed a significant main effect of BMI on overall task performance ($\chi^2$ = 16.80, df = 1, $p_{corrected}$<0.001), such that BMI was negatively associated with accuracy (OR = 0.84, CI = 0.78–0.91, see *Figure 1*). Against our main hypothesis, however, there was no difference in this effect between the WM conditions: the two-way interaction between BMI and condition was insignificant ($\chi^2$ = 2.66, df = 3, $p_{corrected}$>1), indicating no evidence for BMI to have different effects across our WM conditions. We found significant main effects of IQ, gender, tiredness, and concentration (all corrected p-values<0.008). As expected, IQ and concentration were positively associated with task performance ($OR_{IQ}$ = 1.24, $CI_{IQ}$ = 1.14–1.35; $OR_{concentration}$ = 1.30, $CI_{concentration}$ = 1.20–1.41), while tiredness predicted task performance in a negative manner (OR = 0.87, CI = 0.80–0.95). Males performed worse than females on the task (OR = 0.86, CI = 0.82–0.97). Please refer to *Table 2* the full model output displaying the original, uncorrected p-values.

When investigating the interactive effects of COMT and Taq1A on BMI-dependent WM gating (model 2), results reveal that the four-way interaction of BMI × condition × COMT × Taq1A was non-significant ($\chi^2$ = 4.09, df = 6, $p_{corrected}$>1). This indicates that the two SNPs did not have the expected differential effects on WM gating. There were no main effects of COMT ($\chi^2$ = 0.159, df=2, $p_{corrected}$>1)

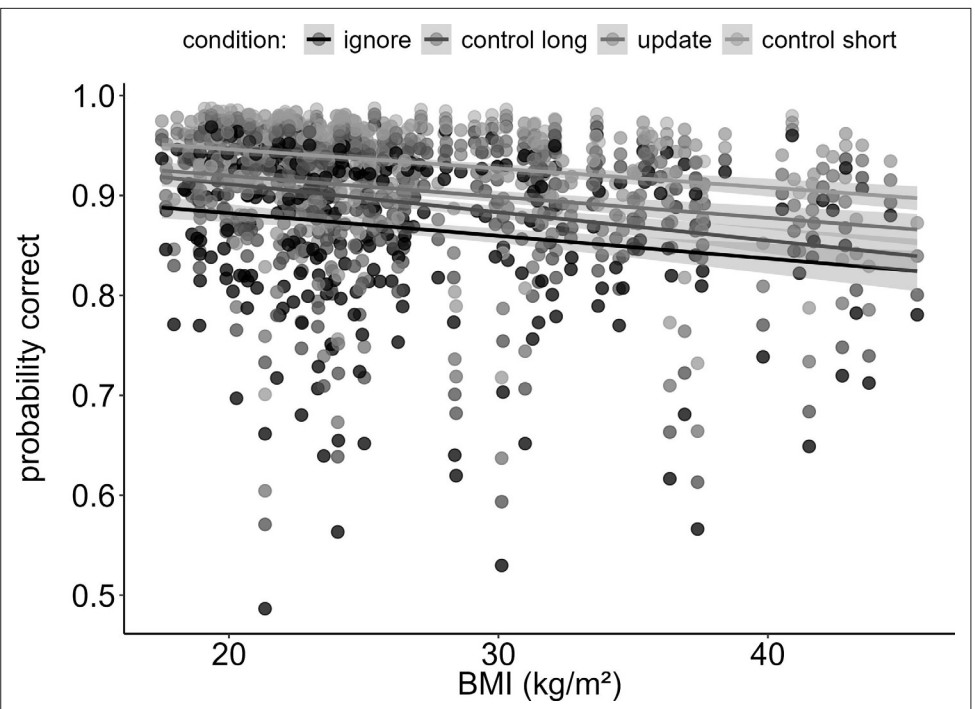

**Figure 1.** Main effect of BMI on working memory performance (model 1). Increasing BMI was associated with worse performance ($p_{corrected} < 0.001$, OR = 0.84). This trend was similar for all four conditions, as there was no interaction between BMI and condition ($p_{corrected} > 1$). Shaded areas represent the 95% confidence intervals. N = 320.

or Taq1A ($\chi^2 = 1.13$, df=1, $p_{corrected}>1$), and all other two- or three-way interactions involving COMT were insignificant (all corrected p-values>0.34). Interestingly, however, we could observe a significant three-way interaction between Taq1A genotype, BMI, and condition ($\chi^2 = 12.40$, df=3, $p_{corrected}=0.024$), indicating that Taq1A genotype might moderate BMI-dependent effects on WM gating. Please refer to *Table 3* for the full output of the final model displaying the original, uncorrected p-values.

To further investigate the significant Taq1A × condition × BMI interaction, we ran simple effects analyses, testing the Taq1A-BMI interaction separately for each condition. These analyses showed that the BMI-genotype interaction was significant in the update condition (p=0.002), but not in the other three conditions (all p-values>0.079), suggesting that the effect was specific to updating and hence

**Table 2.** Full output for the model investigating the BMI-condition interaction (model 1).

|  | Chisq | Df | Pr(>Chisq) |
| --- | --- | --- | --- |
| (Intercept) | 3623.78 | 1 | <0.001 |
| condition | 282.00 | 3 | <0.001 |
| zBMI | 16.80 | 1 | <0.001 |
| zIQ | 25.10 | 1 | <0.001 |
| Gender | 10.50 | 1 | 0.001 |
| zWM_tired | 36.00 | 1 | <0.001 |
| zWM_conc | 9.19 | 1 | 0.002 |
| condition:zBMI | 2.66 | 3 | 0.447 |

N = 320; Marginal $R^2$/conditional $R^2$ = 0.069/0.172.

Note: uncorrected p-values are displayed.

**Table 3.** Full output for the model investigating the COMT-Taq1A-BMI-condition interaction (model 2).

| | Chisq | Df | Pr(>Chisq) |
|---|---|---|---|
| (Intercept) | 3228.77 | 1 | <0.001 |
| condition | 212.81 | 3 | <0.001 |
| COMT | 0.16 | 2 | 0.923 |
| Taq1A | 1.13 | 1 | 0.288 |
| zBMI | 22.15 | 1 | <0.001 |
| zIQ | 24.09 | 1 | <0.001 |
| Gender | 7.53 | 1 | 0.006 |
| zWM_tired | 12.39 | 1 | <0.001 |
| zWM_conc | 30.80 | 1 | <0.001 |
| condition:COMT | 10.30 | 6 | 0.113 |
| condition:Taq1A | 4.69 | 3 | 0.196 |
| COMT:Taq1A | 2.47 | 2 | 0.291 |
| condition:zBMI | 3.49 | 3 | 0.322 |
| COMT:zBMI | 0.86 | 2 | 0.650 |
| Taq1A:zBMI | 2.98 | 1 | 0.085 |
| condition:COMT:Taq1A | 2.09 | 6 | 0.911 |
| condition:COMT:zBMI | 6.29 | 6 | 0.391 |
| condition:Taq1A:zBMI | 12.40 | 3 | 0.006 |
| COMT:Taq1A:zBMI | 3.68 | 2 | 0.159 |
| condition:COMT:Taq1A:zBMI | 4.09 | 6 | 0.665 |

N=318.Marginal $R^2$/conditional $R^2$ = 0.076/0.173.
Note: uncorrected p-values are displayed.

might drive the observed overall three-way interaction (*Figure 2*). Further post hoc examination of the effects on updating revealed that, the association between BMI and performance was significant for A1-carriers (95% CIs: –0.488 to –0.190), with 33.9% lower probability to score correctly per unit change in BMI, but non-significant for non-A1-carriers (95% CIs: –0.153 to 0.129; 1.22% lower probability). Interestingly, compared to all other conditions, in the update condition, the negative association between BMI and task performance was weakest for non-A1-carriers (estimate = –0.012, SE = 0.072, but strongest for A1-carriers estimate = –0.339, SE = 0.076; see *Figure 2* and *Supplementary file 1A*), emphasizing that genotype impacts this condition the most. To further check if this difference in slope was statistically significant across conditions, we stratified the sample into Taq1A subgroups (A1+ vs. A1-) and assessed whether BMI affected task performance differently across conditions separately for each subgroup. This analysis revealed no significant difference in the relationship between BMI and task performance across conditions among A1+ individuals ($p_{BMI*condition}$=0.219). However, within the A1- subgroup, a significant interaction effect between BMI and condition emerged ($p_{BMI*condition}$=0.049). Collectively, these findings suggest that the absence of the A1-allele is linked to improved task performance, particularly in the context of updating, where it seems to mitigate the otherwise negative effects of BMI.

In order to determine whether our results stemmed from mere match/non-match response biases or from proper ignoring/updating, we conducted a follow-up analysis, investigating the effects of the probe type presented at the end of each trial. The probe could either be the target item, a completely novel item, or a distractor item, meaning that the probe was one of the items that had to be encoded initially, but then be overwritten. Thus, a distractor probe measures the cognitive

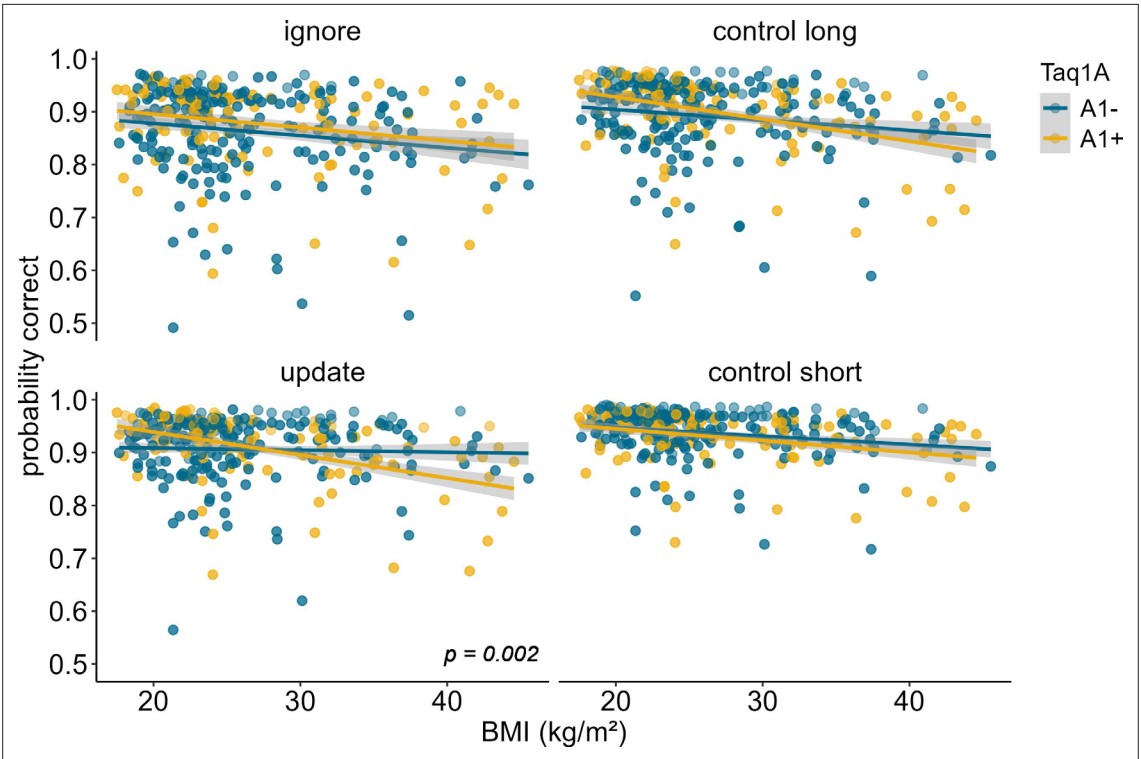

**Figure 2.** Interaction of Taq1A genotype, BMI, and condition on working memory performance (model 2). The two-way interaction of Taq1A and BMI was significant in the update condition only (p = 0.002). In this condition, carrying the A-allele led to a 33.9% decrease in performance with each increasing unit of BMI (SE = 7.58), while there was only a 1.22% (SE = 7.21%) decrease in non-carriers. Shaded areas represent the 95% confidence intervals. N = 318.

challenge of updating in its most exact form, while a target or novel item primarily assays match/non-match responses. For this analysis, we thus subset our data, including updating trials only, and re-ran our model augmented with the factor probe type. Results showed a significant main effect of probe type ($\chi^2$ = 94.11, df = 2, p<0.001). Trials where the probe was a distractor were the hardest (mean accuracy = 86.44%), followed by target probe trials (mean accuracy = 91.58%), and novel probe trials (94.96%). The three-way interaction between probe type, BMI, and Taq1A genotype was not significant ($\chi^2$ = 1.645, df = 2, p = 0.439), indicating that the probe type did not affect the BMI-Taq1A interaction in updating trials. High-BMI A-allele carriers were worse than non-carriers in all three probe trial types similarly. However, this pattern was most pronounced in the distractor condition (see *Figure 3*).

## DARPP-32 genotype moderates the association between BMI and WM updating

Investigation of the effects of DARPP-32 on BMI-dependent WM gating revealed a similar picture to Taq1A. We found a significant three-way interaction of DARPP-32, BMI, and condition ($\chi^2$ = 20.21, df = 3, $p_{corrected}$<0.001), such that DARPP-32 interacted with BMI in the update condition only ($p_{post\ hoc}$=0.006). Please refer to *Table 4* for the full output of the final model displaying the original, uncorrected p-values. Once more, further examination of the observed DARPP-32, BMI, and condition interaction showed that, in the update condition, the negative association between BMI and task performance was weakest and non-significant for A/A (estimate = –0.044, SE = 0.066; 95% CIs: –0.174–0.086), but strongest and significant for G-carrying individuals (estimate = –0.324, SE = 0.079; 95% CIs: –0.478 to – 0.170). See *Supplementary file 1B* and *Figure 4*. Splitting the sample into DARPP subgroups (A/A vs. G-carrier) revealed that in both subgroups, there was significant interaction effect of BMI and condition on task performance ($p_{A/A}$=0.034, $p_{G-carrier}$=0.003). In the case of DARPP, it hence appears that carrying the disadvantageous G-allele could exacerbate the negative effects of BMI, while the more advantageous allele (A/A) might mitigate them – once again particularly in the context of updating. Again, post hoc analyses investigating the effect of probe type, showed

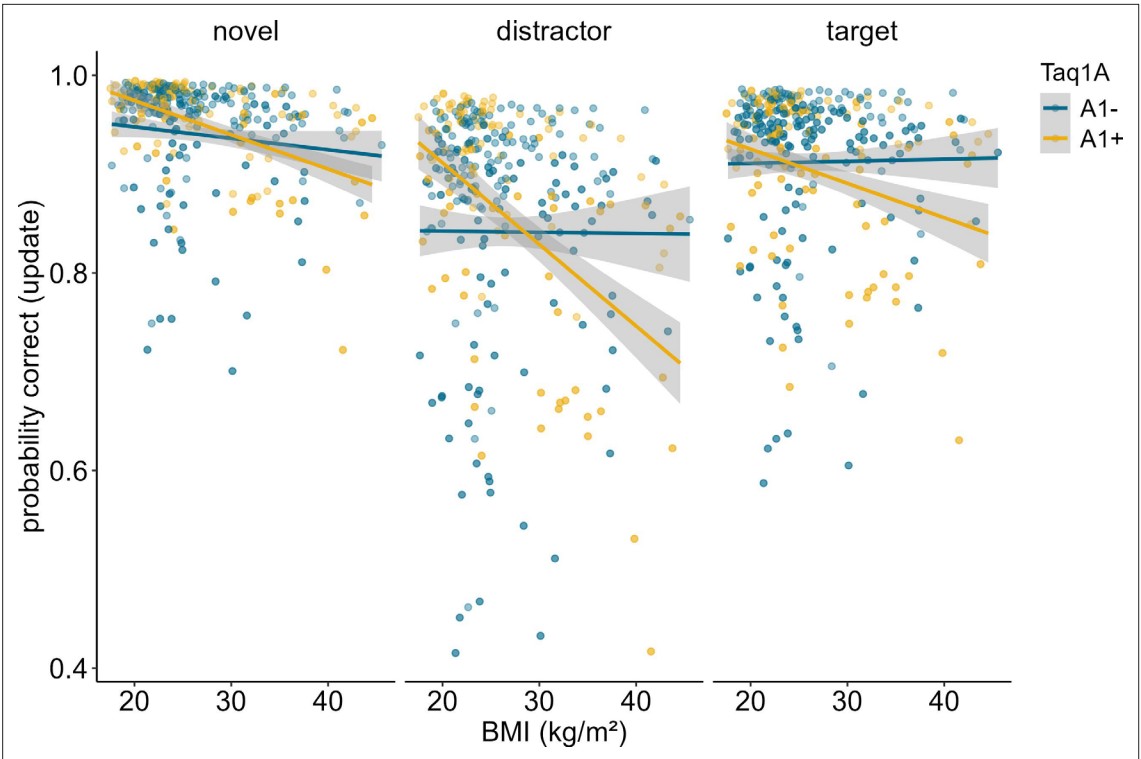

**Figure 3.** Interaction of Taq1A genotype, BMI, and probe type in updating trials only. There was no significant three-way interaction between probe type, BMI and Taq1A (p = 0.439). The BMI - Taq1A interaction was in a similar direction in all trials. There was a significant main effect of probe type (p < 0.001). Trials where the probe was a distractor had lowest probability to be correct. Shaded areas represent the 95% confidence intervals.

that this effect was strongest and significant in distractor (p = 0.046) and target (p = 0.008) trials, but not in trials where the probe was a novel item (p = 0.242, see *Figure 5*). Furthermore, there was a significant main effect of probe type (p < 0.001), with distractor trials having the lowest overall probability to be correct.

**Table 4.** Full output for the model investigating the DARPP-BMI-condition effect (model 3).

|  | Chisq | Df | Pr(>Chisq) |
|---|---|---|---|
| (Intercept) | 3511.81 | 1 | <0.001 |
| DARPP | 0.03 | 1 | 0.853 |
| zBMI | 17.18 | 1 | <0.001 |
| condition | 274.62 | 3 | <0.001 |
| zIQ | 25.10 | 1 | <0.001 |
| zWM_conc | 35.27 | 1 | <0.001 |
| zWM_tired | 10.52 | 1 | 0.001 |
| Gender | 9.17 | 1 | 0.002 |
| DARPP:zBMI | 0.18 | 1 | 0.668 |
| DARPP:condition | 1.00 | 3 | 0.801 |
| BMI:condition | 3.61 | 3 | 0.307 |
| DARPP:BMI:condition | 20.21 | 3 | <0.001 |

N = 320. Marginal $R^2$/conditional $R^2$ = 0.071/0.173.

Note: uncorrected p-values are displayed.

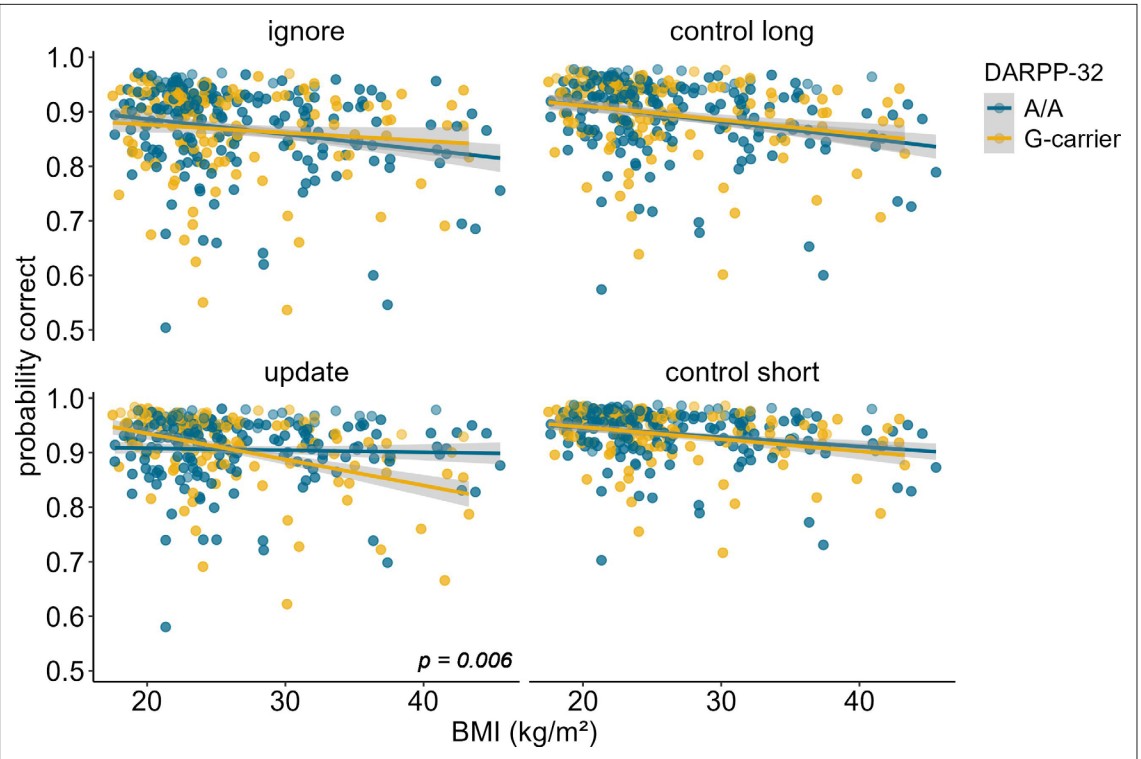

**Figure 4.** Interaction of DARPP-32, BMI, and condition on working memory performance (model 3). The two-way interaction of DARPP-32 and BMI was significant in the update condition only (p = 0.006). In this condition, carrying the G-allele led to a 32.4% decrease in performance with each increasing unit of BMI (SE = 7.86), while there was only a 4.39% (SE = 6.61) decrease in A/A homozygots. Shaded areas represent the 95% confidence intervals. N = 320.

### No association of C957T with BMI-dependent WM gating

Our analysis revealed no significant main effect of the C957T polymorphism ($\chi^2$ = 0.03, df = 1, $p_{corrected}$>1). All other main effects stayed significant (all corrected p<0.012), except for the effect of BMI ($\chi^2$ = 3.49, df = 1, $p_{corrected}$=0.247). Furthermore, we found no substantial evidence for two- or three-way interactions involving the C957T polymorphism (all corrected p>0.186), suggesting that C957T does not significantly interact with BMI or one of our WM conditions. See *Table 5* for the full model output with original uncorrected p-values. Because the main effect of BMI dissipated when including C957T in the model, we ran an additional exploratory analysis to check whether this polymorphism directly related to BMI. A separate lm() model, predicting BMI by C957T, showed no association between the two (p=0.2432), indicating that the BMI effect is probably not masked by the presence of the C957T polymorphism.

### BMI-dependent alterations in WM gating are associated with peripheral dopamine synthesis capacity

When investigating potential influences of dopamine changes on the system level (model 5), we found a significant three-way interaction between amino acid ratio, BMI, and condition ($\chi^2$ = 10.88, df = 3, $p_{corrected}$<0.049). Post hoc simple effects analyses suggested that this interaction seems to be driven by differential performance specifically in update vs. ignore ($\chi 2$ = 5.57, df = 1, p=0.018). As BMI increases, higher ratios of amino acids promote better performance in updating, but worse performance in ignoring (see *Figure 6*, upper panel). All other comparisons (update vs. control short; ignore vs. control long; control long vs. control short, update vs. control long, ignore vs. control short) did not yield significant differential relationships between amino acid ratio and BMI (all p-values>0.168). The main effects of BMI, condition, and amino acid ratio were insignificant (all $p_{corrected}$>1). The main effect of z-IQ ($\chi^2$ = 11.64, df = 1, $p_{corrected}$=0.002) and z-concentration ($\chi^2$ = 18.60, df = 1, $p_{corrected}$<0.001) were significant, both relating positively to performance ($OR_{IQ}$ = 1.28, $OR_{concentration}$ = 1.33).

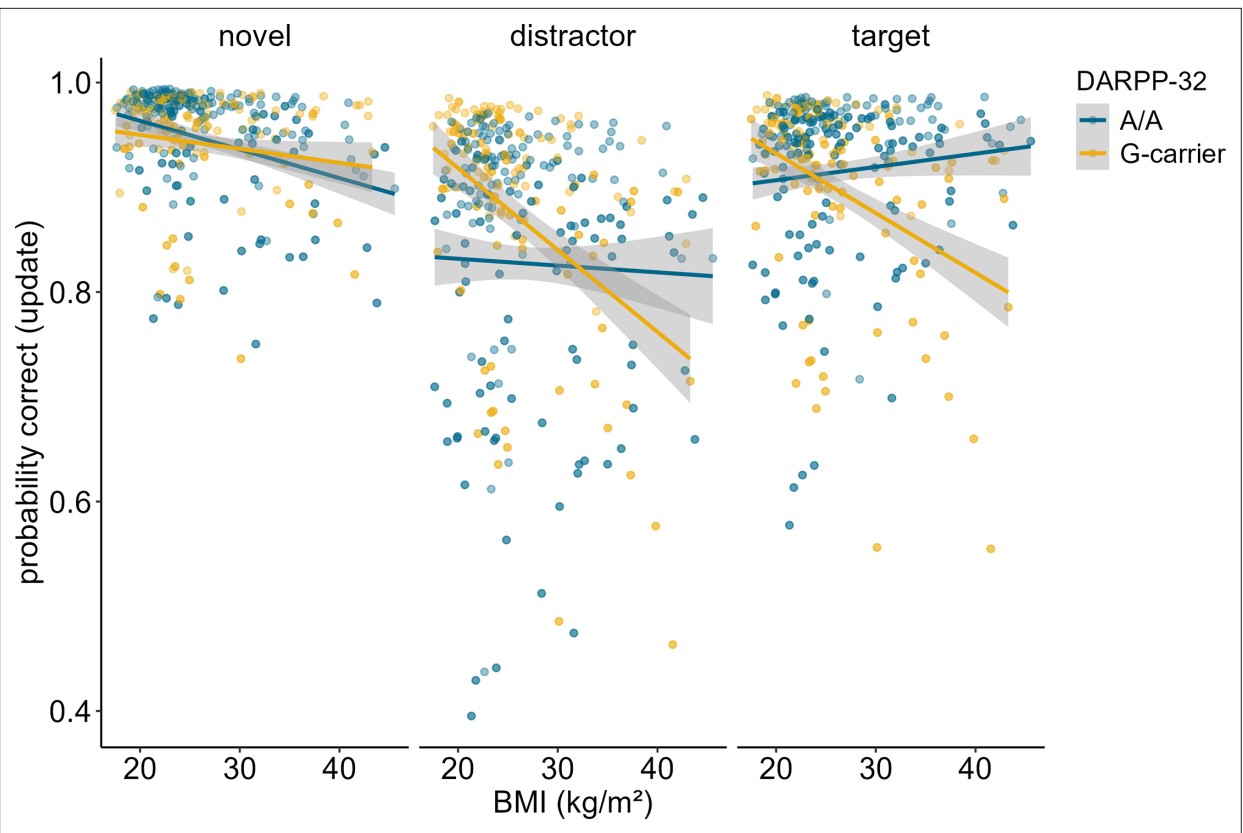

**Figure 5.** Interaction of dopamine and cAMP-regulated neuronal phosphoprotein (DARPP)-32 genotype, body mass index (BMI), and probe type for update trials only. There was a significant three-way interaction between probe type, BMI, and DARPP-32 (p = 0.005). Post hoc analyses showed that the BMI-DARPP interaction was significant in distractor (p=0.046) and target (p=0.008) trials, but not in trials where the probe was a novel item (p=0.242). There was a significant main effect of probe type (p<0.001). Trials where the probe was a distractor had lowest probability to be correct. Shaded areas represent the 95% confidence intervals.

**Table 5.** Full output for the model investigating the C957T-BMI-condition effect (model 4).

|  | Chisq | Df | Pr(>Chisq) |
|---|---|---|---|
| (Intercept) | 328.55 | 1 | <0.001 |
| C957T | 0.03 | 1 | 0.859 |
| zBMI | 3.49 | 1 | 0.062 |
| condition | 48.06 | 3 | <0.001 |
| zIQ | 25.30 | 1 | <0.001 |
| zWM_conc | 33.66 | 1 | <0.001 |
| zWM_tired | 10.54 | 1 | 0.001 |
| Gender | 8.85 | 1 | 0.003 |
| C957T:zBMI | 0.36 | 1 | 0.548 |
| C957T:condition | 7.97 | 3 | 0.047 |
| BMI:condition | 0.31 | 3 | 0.958 |
| C957T:BMI:condition | 0.07 | 3 | 0.995 |

N=318. Marginal $R^2$/conditional $R^2$ = 0.070/0.171.
Note: uncorrected p-values are displayed.

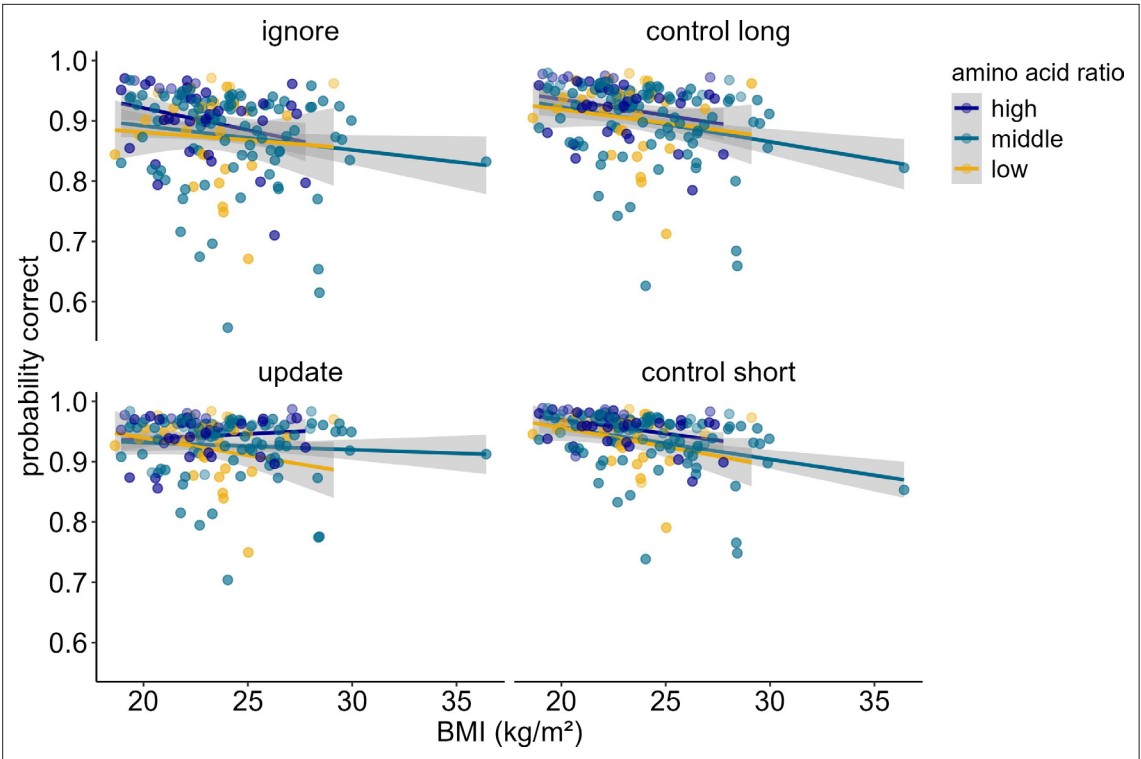

**Figure 6.** Interaction of Amino Acid Ratio, BMI and condition (model 5). For illustration purposes, amino acid ratio was artificially grouped into high, middle, and low. The difference in condition ($p_{\text{ignore vs. update}}$ < 0.001) becomes especially apparent when looking at individuals with high amino acid ratios: With each increasing unit of BMI, performance gets worse in ignore, but better in update. There were no significant differences in the relationship of amino acid ratio and BMI comparing all other conditions against each other (all p > 0.168). Shaded areas represent the 95% confidence intervals. N = 160.

**Table 6.** Full output for the final model investigating the amino acid ratio-BMI-condition effect (model 5).

|  | Chisq | Df | Pr(>Chisq) |
|---|---|---|---|
| (Intercept) | 13.31 | 1 | <0.001 |
| AAratio | 0.58 | 1 | 0.444 |
| zBMI | 0.98 | 1 | 0.321 |
| condition | 3.43 | 3 | 0.330 |
| zIQ | 11.64 | 1 | <0.001 |
| zWM_conc | 18.60 | 1 | <0.001 |
| Gender | 5.08 | 1 | 0.024 |
| AAratio:zBMI | 0.32 | 1 | 0.570 |
| AAratio:condition | 8.69 | 3 | 0.034 |
| BMI:condition | 9.80 | 3 | 0.020 |
| AAratio:BMI:condition | 10.88 | 3 | 0.012 |

N = 160. Marginal $R^2$/conditional $R^2$ = 0.068/0.170.
Note: uncorrected p-values are displayed.

The interactions between BMI and condition ($\chi^2$ = 9.80, df = 3, $p_{corrected}$=0.081), and between amino acid ratio and condition ($\chi^2$ = 8.69, df = 3, $p_{corrected}$=0.135) were not significant. BMI and amino acid ratio showed no significant two-way interaction ($\chi^2$ = 0.322, df = 1, $p_{corrected}$>1). Please refer to *Table 6* for the full model output with original uncorrected p-values. Because there was an extreme BMI data point, we re-ran the model excluding this data point to check whether the results still hold. The three-way interaction between amino acid ratio, BMI, and condition became trend-significant ($p_{corrected}$=0.064, see *Supplementary file 1C* for a full model output with original uncorrected p-values).

## Further exploratory analyses on multiple gene-gene and gene-amino acid ratio interaction effects

Because possible interactions of all SNPs presented in this paper have been shown (*Zmigrod and Robbins, 2021*; *Frank and Hutchison, 2009*; *Smith et al., 2014*; *Xu et al., 2007*), we additionally checked for multiple gene-gene interactions in a highly explorative manner. First, because specific C957T and Taq1A interaction effects on WM have been reported (*Frank and Hutchison, 2009*), we check for a BMI- and condition-dependent interaction of these two SNPs. There were no significant interaction effects (all $p_{uncorrected}$>0.138). Furthermore, we ran a model where we investigated the interaction of C957T and COMT, as also these two SNPs have been shown to interact (*Xu et al., 2007*). There was a significant condition:C957T:COMT:BMI interaction effect ($p_{uncorrected}$=0.026). Last but not least, since our initial analyses showed significant effects for each, we also looked at how amino acid ratio and Taq1A, and amino acid and DARPP would interact. For the latter, we found a significant condition:AAratio:DARPP:BMI interaction effect ($p_{uncorrected}$=0.03). Full outputs for all models can be found in *Supplementary file 1D-G* .

## Discussion

The present investigation sought to evaluate whether obesity might be associated with impairments in WM gating. Consistent with previous literature (*Yang et al., 2020*; *Yang et al., 2019*; *Yang et al., 2018*; *Gonzales et al., 2010*; *Coppin et al., 2014*; *Hartmann et al., 2023*), we found evidence for impairments in overall WM in individuals with a high BMI. Yet, we could not observe the expected interaction of BMI and condition, indicating no specific associations between BMI and WM gating. Interestingly, however, distinct effects of BMI on gating became apparent when taking into account potential changes in inherent dopamine signaling. Specifically, Taq1A and DARPP-32 particularly affected performance in the updating condition. Against our expectation, however, we did not find evidence for an interaction effect of COMT and Taq1A on BMI-dependent WM gating, nor did we find any effects of C957T.

## Selective BMI-genotype effects on WM updating

Our findings are partially in line with our hypothesis. While we did observe the expected worsening of updating WM contents in individuals with a high BMI, this effect was not exclusive to updating. Only when participants – along with BMI – were categorized based on certain putative dopamine-signaling characteristics, distinct effects on updating became apparent. This finding is compelling as it demonstrates a rarely observed selective effect. Notably, it were the Taq1A and DARPP-32 SNPs that selectively modulated WM updating in a BMI-dependent manner. Intriguingly, both of these SNPs are associated with, predominantly, striatal dopamine signaling (*Hemmings and Greengard, 1986*; *Meyer-Lindenberg et al., 2007*; *Gluskin and Mickey, 2016*), implying a targeted modulation of processes occurring within the striatum. Both, the A-allele (Taq1A) and the G-allele (DARPP-32), have previously been considered risk alleles for various conditions and behaviors involving maladaptive cognitive flexibility, such as addiction (*Smith et al., 2008*; *Munafò et al., 2007*; *Deng et al., 2015*), schizophrenia (*Meyer-Lindenberg et al., 2007*; *González-Castro et al., 2016*; *Albert et al., 2002*), or impaired reinforcement learning (*Frank et al., 2007*; *Doll et al., 2011*). Noteworthy, our data revealed that differences in updating appeared to be driven by the non-risk allele groups. Despite increasing BMI, performance remained stable. This pattern suggests that possessing the more advantageous genotype could potentially mitigate the generally negative effects of a high BMI on WM updating. Moreover, in the normal-weight BMI range, carriers of a risk allele (in both, Taq1A and/or DARPP) slightly outperformed their non-risk allele carrying counterparts. This is especially intriguing as it

emphasizes that carrying a 'risk allele' can in fact be advantageous under certain cognitive demands – a claim that has also been put forward by, e.g., *Stelzel et al., 2010*. Finally, the effects of genotype were particularly pronounced in trials where the probe was a distractor, suggesting that the effect is primarily due to 'real' updating, i.e., when initially encoded items need overwriting, as opposed to simple match/non-match responses (as in novel vs. target probe items).

## Potential mechanistic accounts

Mechanistically, our findings are potentially due to differential go/no-go path activation in the basal ganglia – pathways that are crucially involved in governing WM gating. In essence, the D1 pathway modulates the 'go' signaling responsible for updating, while the D2 pathway facilitates 'no-go' signaling crucial for distractor-resistant maintenance (*Frank and O'Reilly, 2006*). Considering Taq1A, evidence points at increased striatal dopamine synthesis and corresponding increases in striatal BOLD signals in A-carriers compared to non-carriers (*Laakso et al., 2005*; *Stelzel et al., 2010*). These findings suggest that the phasic dopamine signal needed to trigger 'go' (i.e. updating), might be enhanced in A-carriers. This aligns with the idea that A-carriers, who possess fewer D2 receptors (*Thompson et al., 1997*; *Pohjalainen et al., 1998*; *Jönsson et al., 1999*), fall more within the ambit of the D1/go-dominant regime (*Klein et al., 2007*). Our data support this speculation by revealing slightly better updating performance in A-carriers in the normal-weight BMI range. However, as BMI increases, the possession of a greater D2 receptor density seems to become advantageous, as evidenced by the lack of a negative correlation between BMI and updating performance in non-A carriers. We speculate that this phenomenon could potentially be attributed to the compensating effects of this genotype. While individuals with fewer D2 receptors (A1+) may have quicker saturation of receptors regardless of dopamine levels, in those with more D2 receptors (A1-) saturation may be slower. This could contribute to a more finely tuned balance between 'go' and 'no-go' signaling, despite potential alterations in dopamine tone in obesity (*Horstmann et al., 2015*; but also see *Darcey et al., 2023* or *Janssen and Horstmann, 2022*). Clearly, the current data cannot provide empirical evidence for these speculations, and further discrete research is needed to establish firm conclusions. Regarding DARPP, we found that carrying the G-allele significantly exacerbated the negative effects of BMI, while the more advantageous allele (A/A) mitigated them, once again particularly in the context of updating. Interestingly, the G-allele has been associated with reduced striatal D1 efficacy (*Meyer-Lindenberg et al., 2007*). Moreover, *Frank et al., 2007*, and *Doll et al., 2011*, showed that carrying a G-allele was associated with worse go-learning – a process requiring activation of the same go-pathway that is likely to be activated during updating of WM contents. Similarly, *Frank and Hutchison, 2009*, demonstrate that the G-carrier group compared to the A/A allele group displayed worse approach learning – again a process relying on go-path activation. However, there were no effects of DARPP on no-go learning (*Frank et al., 2007*), which requires D2-mediated no-go path activation. Our results hence broadly align with the literature and suggest that, particularly, markers of striatal go-signaling modulate BMI-dependent effects on WM updating. Collectively, our observations hint at the potential of advantageous genotypes to moderate the adverse impacts of high BMI on cognitive functions.

## Possible accounts for the absence of COMT and C957T effects

Considering that according to the prevailing models, PFC, and striatum interact to foster effective WM gating (*Cools and D'Esposito, 2011*), the question arises as to why we could not observe the expected COMT-Taq1A interaction on BMI-dependent WM gating. We posit several explanations for the absence of the anticipated interaction. First, these polymorphisms may indeed exert a limited interactive effect on WM gating. In line with this notion, prior findings concerning the interactive effects of COMT and Taq1A on WM have yielded contradictory results. For instance, *Garcia-Garcia et al., 2011*, and *Stelzel et al., 2009*, reported patterns of COMT-Taq1A interactions in the context of WM updating that were consistent in terms of direction of effect. In contrast, *Wishart et al., 2011*, observed an opposing interaction pattern, while *Persson and Stenfors, 2018*, did not identify any COMT-Taq1A interaction at all. All these studies explored genotype interactions using paradigms that either assessed the two memory processes separately (*Garcia-Garcia et al., 2011*; *Wishart et al., 2011*; *Persson and Stenfors, 2018*) or in a manner that they were not distinctly discernible (*Berry-hill et al., 2013*). None of them examined the comprehensive interaction of COMT, Taq1A, and WM updating vs. ignoring within a single paradigm, as we did here. Second, it has recently been debated

whether COMT has a noteworthy effect on cognition. Some meta-analyses find (small) effects (*Barnett et al., 2007*), while others don't (*Geller et al., 2017*; *Barnett et al., 2008*; also see *Goldman et al., 2009*; *Wacker, 2011*; *Barnett et al., 2011*, for a discussion on the meta-analysis from *Barnett et al., 2008*). Beyond this, the absence of significant effects related to COMT could further be interpreted as underscoring the selectiveness of our observed effects. COMT effects are predominantly observed in the PFC (*Mier et al., 2010*; *Egan et al., 2001*; *Käenmäki et al., 2010*) and rather tied to maintenance of WM contents (*Nolan et al., 2004*; *Rosa et al., 2010*). This lends weight to the interpretation that distractor-resistant maintenance, or prefrontal processes, remain unaffected by BMI. In a similar vein, also our findings concerning the C957T polymorphism bolster the selective nature of our findings. Much like COMT, this polymorphism is presumably more involved in WM maintenance, i.e., prefrontal-related functioning. Supporting this notion, *Xu et al., 2007*, found an association between the C957T polymorphism and specifically maintenance of (phonological and serial) information, but not with other tasks requiring updating. Furthermore, this polymorphism has also been associated with D2 binding potential in extrastriatal regions (*Hirvonen et al., 2009*) and greater WM-related activity in PFC (*Li et al., 2019*). Lastly, the results with respect to C957T involvement in striatal-dependent cognition are mixed (see, e.g., discussion part in *Baker et al., 2013*), indicating that C957T may not be a good candidate for influencing striatal-dependent processes.

## Selective modulation of WM gating: system-level dopamine versus genetic profiles

Our BMI-gene findings show a selective modulation of WM updating, as opposed to the previously observed trade-off between ignoring and updating (e.g. *Fallon et al., 2017b*; see *Cools, 2019*, for an extensive review). To the best of our knowledge, none of the previous studies investigating WM gating in relation to dopamine signaling have found such a selective modulation. We speculate that this is because previous studies looked at broader changes in the dopamine system, i.e., by using drug manipulations or comparing Parkinson's vs. healthy controls, rather than particular genetic profiles. Such broad 'system-level' dopamine changes may impact both PFC-facilitated distracter resistance and striatal-dependent updating. This, in turn, might foster the commonly observed inverted-U-shaped relationship between dopamine and cognition: In cases where baseline dopamine levels are low, a dopamine increase (for instance, through agonists) would enhance ignoring, albeit at the expense of updating. Conversely, at medium baseline dopamine levels an increase would lead to impaired ignoring, potentially benefiting updating (for a more detailed discussion, see *Cools and D'Esposito, 2011*). Indeed, we also see this pattern when looking at system-level dopaminergic changes: depending on BMI, low (or high) peripheral dopamine synthesis capacity (as indicated by blood amino acid ratios) was associated with worsening of distractor-resistant maintenance, while improving updating (or vice versa). It should be noted that the sample for our amino acid analyses was much smaller (N = 160) than the one used for the SNP analyses (N = 320), and the BMI range for this sub-sample was narrower (mean = 23.63, SD = 2.78, min = 18.63, max = 36.42). This was because only two of the three studies had the data on amino acids available. Interestingly, the system-level effect of amino acid ratio becomes visible in a healthy to overweight BMI range, indicating that already small changes in BMI can promote different dopamine-dependent cognitive profiles.

## Strengths and limitations

A major strength of this study is that it was the first to probe gene-gene interactions on a direct WM maintenance and updating comparison. This is a notable advantage, as previous studies have usually examined these aspects in separate paradigms, which might have contributed to the heterogeneous results regarding SNP interaction effects on WM (see above). Another main strength of our study is the sizable sample. However, despite this relatively large size, our sample was still not big enough for systematic and reliable analysis of multiple gene-gene interactions. This would be of interest however, as possible interactions of all SNPs presented in this paper have been shown (*Zmigrod and Robbins, 2021*; *Frank and Hutchison, 2009*; *Smith et al., 2014*; *Xu et al., 2007*). Such analyses would require even larger cohorts, as the effect sizes of single SNPs are usually small. We nevertheless report the outcome of such highly explorative models in our exploratory analyses for the purpose of transparency and to guide future studies. Yet, those results should be interpreted with caution. Furthermore, an additional limitation is that our data is slightly skewed toward participants within the normal BMI

range. The effective sample size to detect meaningful genotype effects (e.g. for COMT or C957T) might thus have been too small, particularly at higher BMI levels. Future studies may address this limitation by recruiting a more balanced sample, including more individuals with higher BMI. Additionally, the correlational nature of our findings highlights the need for more direct experimental manipulations of dopaminergic processes in obesity. Previous studies have established a causal link between dopamine and WM gating through drug manipulations (*Fallon et al., 2017b*). Applying a similar approach to an obese sample could help establish a clearer causal link between dopamine activity and WM gating in the context of obesity. Lastly, the sample used for this study was very heterogeneous, as it was pooled from three separate studies. The BMI distribution, for instance, was significantly different depending on gender. BMI was higher in females. This was because females were over-represented in the BEDOB study, which had the largest BMI range (refer to *Table 1*). Although we ran control analyses to account for this heterogeneity, we cannot exclude the possibility that certain properties of the data distribution could have influenced our results.

## Overall conclusions

Overall, our data aligns with previous evidence for WM impairments in obesity. However, selective effects of BMI on WM gating – specifically updating – become visible only when accounting for genetic markers of striatal dopamine transmission. This level of specificity adds a new nuance to existing research as it demonstrates condition-specific effects of BMI on WM. Previous research, that generally utilized drug manipulations, consistently demonstrated system-level modulations, leading to a trade-off between ignoring and updating information. While we also observe this trade-off when examining more comprehensive system-level relationships (i.e. blood amino acid ratio), the specificity of our SNP-related findings to updating sets them apart from previous studies. Our results hence pave the way for new individualized treatments for obesity, as they highlight the potential of advantageous genotypes to mitigate the adverse effects of high BMI on cognitive functions that require updating of information, and suggest that previously documented deficits in reward learning, which partially rely on information updating, could potentially be targeted more specifically when taking genotypes into account.

# Materials and methods
## Participants

The data used in this study were collected in the scope of three separate pre-registered cross-sectional studies, which are all part of a larger line of research in the O'Brain Lab: GREADT (see https://osf.io/w9e5y), BEDOB (see https://osf.io/fyn6q), and WORMCRI (see https://osf.io/zdmkx). The studies were conducted in compliance with the principles of the Declaration of Helsinki and authorized by the Ethics Committee of the Medical Faculty at the University of Leipzig (400/18-ek; 385/17-ek; 172/19-ek). All participants provided written informed consent before participation and were compensated for their time. Prior to participation, participants were screened for a history of clinical drug or alcohol abuse, neurological or psychiatric disorders, and first-degree relative history of neurological or psychiatric disorders. Symptoms of depression were assessed via a screening interview using the Structured Clinical Interview for DSM-IV (SCID, *Wittchen, 1997*; in BEDOB & WORMCRI) or Beck Depression Inventory (BDI, *Beck et al., 1996*; in GREADT).

## Study design

All measures relevant to the present study were collected in a comparable manner. In all studies, participants were first asked to come to the lab for a screening session where in- and exclusion criteria were checked. Weight and height were measured to calculate BMI. After inclusion, blood samples were taken from the participants to assess COMT Val[158]Met, Taq1A, C957T, and DARPP-32 genotypes. Analysis of these SNPs was performed in the laboratory for 'Adiposity and diabetes genetics' at the Medical Research Center, University Leipzig, Leipzig, Germany. In WORMCRI and GREADT, we also took serum blood samples in order to extract information on the amino acid profiles. Participants, therefore, came overnight-fasted for these two studies. Serum blood samples were analyzed at the 'Institut für Laboratoriumsmedizin, Klinische Chemie und Molekulare Diagnostik (ILM)' Universitätsklinikum Leipzig, Germany. After the blood draw, participants did a number of neuropsychological tests

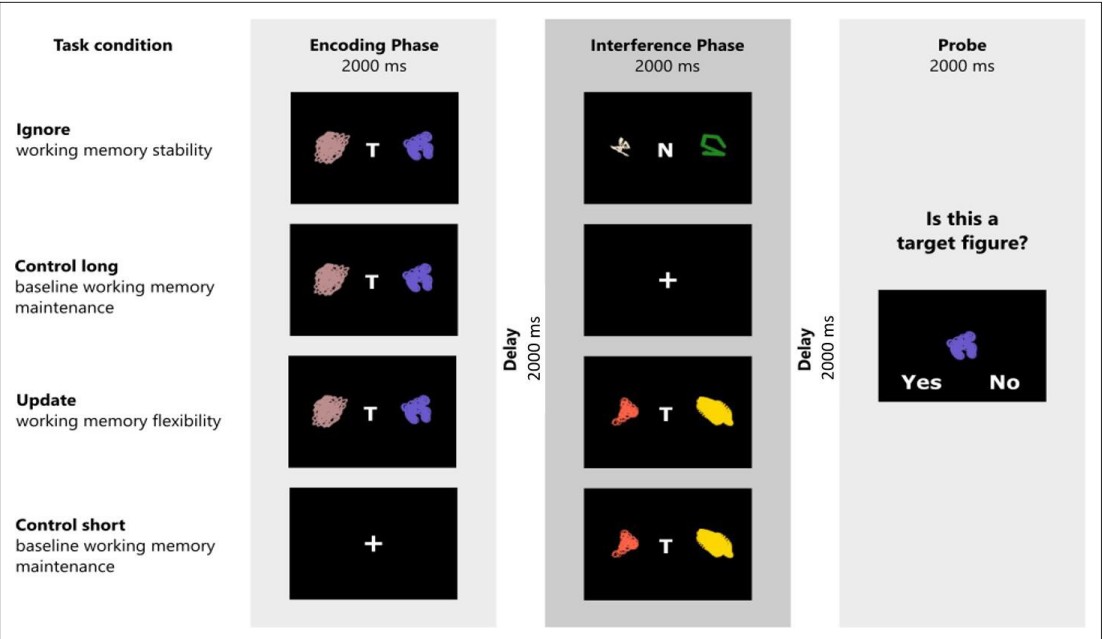

**Figure 7.** Schematic illustration of the task structure and experimental conditions. The task consists of three task phases. In the encoding phase, participants have to remember two target stimuli (signaled by the letter "T"), or are presented with a centered cross (short control trials). In the interference phase, participants either have to ignore two non-target stimuli (ignore trials; signaled by the letter "N") or allow two new stimuli (again marked by a "T") to replace the previously remembered target stimuli (update trials). No-interference trials (short and long control) do not require any manipulations in the interference phase. At the end of each trial, participants evaluate whether a presented figure was a target figure or not. Figure reused from Hartmann et al. (2023) with permission.

among which were the digit span task (*Hilbert et al., 2014*; *Wechsler, 2008*; assessing baseline WM), and a proxy for IQ in BEDOB: 'Wortschatztest' (*Schmidt and Metzler, 1992*; assessing verbal IQ); in GREADT and WORMCRI: 'Wiener Matrizen Test' (*Formann et al., 2011*). After that, participants filled in several questionnaires, of which the DFSQ (*Francis and Stevenson, 2013*; *Fromm and Horstmann, 2019*; assessing eating behavior), was subject to all three studies. On the second test day, participants completed the WM task (described below), either during fMRI (GREADT and BEDOB) or during EEG (WORMCRI). After completion of the task, all participants were asked to indicate the level of tiredness and concentration they felt during the task on a 10-point Likert scale. For a more detailed description of each study's design, please refer to the respective pre-registration mentioned above.

## WM task

Participants completed a modified version of a delayed match-to-sample task originally designed by *Fallon and Cools, 2014*. This modified version has already been described in *Hartmann et al., 2023*, and *Herzog et al., 2023*. The task comprises four conditions (*Figure 7*). In the ignore condition, testing distractor-resistant maintenance, participants first have to memorize two target stimuli, signaled by the letter 'T' centered in between the two. Next, they are presented with two new stimuli, this time marked by a centered 'N', indicating non-targets which have to be ignored. After that, participants are presented with a probe stimulus and have to determine whether one of the first two target stimuli matches the presented probe. In the update condition, participants are first shown two target stimuli (centered 'T'). After that, they see a new set of target stimuli (again indicated by a centered 'T'). These two new stimuli replace the previously presented stimuli as the target and thus have to be evaluated for a match when the probe is presented subsequently. The two control conditions do not have any interference and are matched to the temporal delay between encoding the to-be-matched targets and probe. The probe is presented for 2000 ms. The task is separated into four blocks, with each block entailing 8 trails of each condition, interleaved among all blocks. Each block thus consists of 32 trials. The total number of trials in the task amounts to 128. Feedback is presented after each of those blocks. Each trial is separated by a jittered inter-trial interval ranging from 2000 to 6000 ms. The

stimuli are randomly computer-generated, monochromatic RGB 'spirographs'. The primary outcome measure is accuracy. The total duration of the task is approximately 30 min.

## Statistical analyses of behavioral data

All behavioral analyses were performed in R in RStudio v4.2.2 (*Dreisbach and Fröber, 2019*; *R Development Core Team, 2015*). Given the within-subject design of our study, we used generalized linear mixed models of the '*lme4*' package to analyze the primary outcome measure of the WM task: accuracy. We ran a logistic regression using *glmer*() with a binomial link function. We used trial-by-trial information for each subject with binary coded response (0 = incorrect; 1 = correct). Trials with a reaction time <200 ms and >2000 ms were excluded, as those trials can be considered false alarms and misses, respectively. Trials with a reaction time >2000 ms were excluded, as they reflect misses. To first test our main hypothesis that WM gating is altered, depending on BMI, we built a trial-based regression model including the interaction of the within-subject factor condition (ignore vs. update vs. control long vs. control short) and the continuous between-subject factor BMI. We further probed the influence of several potential covariates: study (GREADT vs. BEDOB vs. WORMCRI), IQ, Age, DFSQ, binge-eating phenotype, tiredness, concentration, and gender. Using the *anova*() function from the '*stats*' package, we compared AIC and BIC (*Akaike, 1979*; *Stone, 1979*) of the full model against multiple simpler version of the model. We found the best-fitting models (lowest AIC and BIC) to include IQ, tiredness, concentration, and gender (see *Supplementary file 1* below). Due to model convergence problems, the continuous predictors BMI, IQ, tiredness, and concentration were z-scored. Furthermore, the model did not converge with a maximal random structure (including the within-subject factor 'condition'). The random structure of the model was thus reduced to include the factor 'subject' only, thereby accounting for the repeated measures taken from each subject. The final model was:

$$accuracy \sim condition * BMI + IQ + tiredness + concentration + gender + (1|subject) \tag{1}$$

To test how BMI-dependent WM gating is moderated by the respective dopamine proxy (SNP or amino acid ratio), we ran four additional models, each including the respective between-subject factor as an additional factor of interest. Model fit was again assessed using AIC and BIC for each model (see *Supplementary file 1H*). Again, due to convergence issues, the random structure of the models included the factor 'subject' only. The final models were:

$$accuracy \sim COMT * Taq1A * condition * BMI + IQ + tiredness + concentration + gender + (1|subject) \tag{2}$$
$$accuracy \sim DARPP * condition * BMI + IQ + tiredness + concentration + gender + (1|subject) \tag{3}$$
$$accuracy \sim C957T * condition * BMI + IQ + tiredness + concentration + gender + (1|subject) \tag{4}$$
$$accuracy \sim aminoacidratio * condition * BMI + IQ + concentration + gender + (1|subject) \tag{5}$$

As we ran four additional models testing similar hypotheses, all main results for these models were corrected for multiple comparisons using Bonferroni correction, i.e., p-values were multiplied by 4. Model outputs were called using the *Anova*() function, from the '*car*' package. Reported odds ratios (OR) are retrieved from exponentiating the log-odds coefficients called with the *summary*() function.

## Acknowledgements

The authors thank Sylvia Stasch, Miriam Huml, Eva Burmeister, and Lisa Okhof for helping with recruitment of participants and data collection. Furthermore, we thank Ines Müller from Peter Kovacs lab at the University of Leipzig for the analyses of our blood samples. Last but not least, we thank Susan Prejawa for her assistance in study organization and financial management.

## Additional information

### Funding

| Funder | Grant reference number | Author |
|---|---|---|
| FAZIT Stiftung | | Nadine Herzog |
| "Wiedereinstiegsstipendium" from University of Leipzig | | Nadine Herzog |
| Max Planck Society | | Nadine Herzog |

The funders had no role in study design, data collection and interpretation, or the decision to submit the work for publication.

### Author contributions

Nadine Herzog, Conceptualization, Data curation, Formal analysis, Funding acquisition, Visualization, Writing - original draft, Writing - review and editing; Hendrik Hartmann, Arsene Kanyamibwa, Peter Kovacs, Lorenz Deserno, Sean Fallon, Arno Villringer, Annette Horstmann, Writing - review and editing; Lieneke Katharina Janssen, Conceptualization, Supervision, Writing - review and editing; Maria Waltmann, Data curation, Writing - review and editing

### Author ORCIDs

Nadine Herzog ⓘ https://orcid.org/0000-0002-8346-7153
Arsene Kanyamibwa ⓘ https://orcid.org/0000-0001-6190-4626
Maria Waltmann ⓘ https://orcid.org/0000-0001-7938-6046
Peter Kovacs ⓘ https://orcid.org/0000-0002-0290-5423
Lorenz Deserno ⓘ https://orcid.org/0000-0001-7392-5280

### Ethics

The study was conducted in compliance with the principles of the Declaration of Helsinki and was authorized by the Ethics Committee of the Medical Faculty at the University of Leipzig (172/19-ek; 385/17-ek; 400/18-ek). All participants provided written informed consent.

Reviewer #1 (Public review): https://doi.org/10.7554/eLife.93369.3.sa1
Reviewer #2 (Public review): https://doi.org/10.7554/eLife.93369.3.sa2
Author response https://doi.org/10.7554/eLife.93369.3.sa3

## Additional files

### Supplementary files

• MDAR checklist

• Supplementary file 1. Supplementary tables. (A) Post hoc effects for the interaction of BMI, condition, and Taq1A. (B) Post hoc effects for the interaction of BMI, condition, and DARPP. (C) Full output final model 5 without extreme data point. (D) C957T-Taq1A interaction. (E) C957T-COMT interaction. (F) AminoAcidRatio-Taq1A interaction. (G) AminoAcidRatio-DARPP interaction. (H) Final outputs for the model comparisons for each model.

### Data availability

Data and scripts used for the analysis are available at https://github.com/O-BRAIN/WM_SNP, copy archived at *O-BRAIN, 2023*.

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
