## [Editor Report · eLife assessment]

The present study provides **valuable** evidence on the neurochemical mechanisms underlying working memory in obesity. The authors' approach considering specific working memory operations (maintenance, updating) and putative dopaminergic genes is **solid**, though the inclusion of a more direct measure of dopamine signaling would have strengthened the work.

---

## [Referee Report · Reviewer #1 (Public review)]

Herzog and colleagues investigated the interactions between working memory (WM) task condition (updating, maintenance) and BMI (body-mass-index), while considering selected dopaminergic genes (COMT, Taq1A, C957T, DARPP-32). Emerging evidence suggest that there might be a specific negative association with BMI in the updating but not maintenance condition, with potential bearings to reversal reward learning in obesity. The inclusion of multiple dopaminergic genes is a strength in the present study, considering the complexity of the interactions between tonic and phasic dopamine across the brain that may distinctly associate with the component processes of WM. Here, the finding was that BMI was negatively associated with WM performance regardless of the condition (updating, maintenance), but in models including moderation by either Taq1A or DARPP-32 (but not by COMT and C957T) an interaction by task condition was observed. Furthermore, a two-way interaction effect between BMI and genotype was observed exclusively in the updating condition. These findings are in line with the accounts by which striatal dopamine as reflected by Taq1A and DARPP-32 play an important role in working memory updating, while cortical dopamine as reflected by COMT is mainly associated with maintenance. The authors conclude that the genetic moderation reflects a compound effect of having high BMI and an advantageous allele in Taq1A or DARPP-32 to working memory updating specifically.

These data increment the accumulating evidence that the dopamine system plays an important role in obesity. The result that Taq1A and DARPP-32 moderated the interaction between WM condition and BMI required intricate post hoc analysis to understand the bearings to updating. The authors found that Taq1A or DARPP-32 genotype moderated the negative association between BMI and WM exclusively in update condition (significant two-way interaction effect), suggesting that the BMI-WM associations in other conditions were similar across genotypes. Importantly, visual inspection of the relationship between WM and BMI (Fig 4 & 5) suggests more prevalent positive effects of the putatively advantageous Taq1A-A1 and DARPP-32-AA genotypes to the overall negative relationship between WM and BMI in updating, but not in the other conditions. Given that an overall negative relationship was statistically supported across all conditions (model 1), a plausible interpretation would be that updating condition stands out in terms of a positive moderation by putative advantageous genotypes, rather than compound negative consequences of BMI and genotype in updating. Statistical testing stratified by Taq1A genotype confirmed that the interaction with task condition was driven by the carriers of the advantageous genotype, whereas stratification by DARPP-32 genotype revealed a significant task-condition interaction in both A/A- and G-carriers. Taken together, the present results highlight inter-subject variability in the associations between obesity, dopamine, and working memory, which can sometimes be captured using blood-based dopamine markers. This finding indicates that not all individuals with obesity show the same patterns of dopamine-related alterations and underscores the necessity to address inter-individual variability in future research and treatment efforts.

---

## [Referee Report · Reviewer #2 (Public review)]

Summary:

The authors investigated if obesity is associated with elevated working memory deficits. Prior theorizing would suggest that individuals with a higher BMI would be worse at working memory updating, potentially due to impaired dopaminergic signaling in the striatum. However, the authors find that higher BMI was associated with worse working memory performance, irrespective of having to ignore or update new information. To further explore the putative dopaminergic mechanisms, participants are stratified according to genetic polymorphisms in COMT, Taq1A, DARPP and C957T and the ratio of the amino acids phenylalanine and tyrosine, all implicated in dopamine-signaling. They find that carrying specific alleles of Taq1A and DARPP, but not of COMT and C957T, mitigated the otherwise negative relationship between BMI and working memory for updating, but not for maintenance.

The authors put forward several possible mechanistic explanations of these observations, including imbalances in the striatal go/no-go dopamine pathways. However, only future, more direct measures of dopamine signaling can provide a confirmation of these hypotheses.

Strengths:

Differentiating between working memory maintenance (ignoring) and updating is a powerful way to get a deeper insight into specific working memory deficits in individuals with obesity. This way of assessing working memory could potentially be applied to various populations at risk for cognitive or working memory deficits.

By pooling data from three studies, the authors reached a relatively large sample of 320 participants, which enables the assessment of more subtle effects on working memory, including the differentiation between updating and ignoring.

Working memory gating has long implicated striatal dopamine signaling. This paper shows that a specific combination of a high BMI and specific dopamine-related genotypes can selectively moderate working memory updating. More insight into how these risk factors interact can ultimately lead to more tailor-made treatments.

Weaknesses:

The introduction mentions that specific alleles can alter dopamine signaling in various ways. However, the authors are less clear on how they expect these alterations to subsequently affect working memory updating and maintenance in the current study. While I understand that the complexity of these mechanisms might make it challenging to form specific predictions, it would be helpful if the authors acknowledged this uncertainty and clarified that their analyses are exploratory in nature, and they will therefore refrain from any directional hypotheses regarding the genotypes.

The majority of participants seems to fall within the normal BMI-range, whereas the interaction between BMI and genetic variations or amino acid ratio particularly surfaces at higher BMI. As genetic variations are usually associated with small effect sizes, the effective sample size, although large for a behavioral analysis only, might have been too small to detect meaningful effects of particular alleles of COMT and C957T.

The relationships between genetic variations, BMI and specific disturbances in dopamine signaling are complex, as compensating mechanisms might be at play to mitigate any detrimental effects. Future studies that apply more direct measures or manipulations of dopaminergic processes could therefore aid in mechanistically explaining the results.

---

## [Author Response]

The following is the authors’ response to the original reviews.

**Reviewer #1 (Public Review):**
In particular, theoretical analysis of the extant evidence and formulation of the hypothesis remains elusive in terms of the potential mechanisms of updating/maintaining balance in obesity

We thank the reviewer for their feedback regarding the theoretical analysis and hypothesis formulation in our manuscript. We have attempted to build our hypothesis based on established correlations between dopamine levels and working memory capabilities, as seen in various populations affected by dopaminerelated changes (e.g. Parkinson’s disease (Fallon et al. 2017), older individuals (Podell et al., 2012), or more generally, in individuals with lower dopamine synthesis capacity (Colzato et al., 2013)). Our hypothesis — that individuals with higher BMI might show impaired updating — is an extrapolation from observed patterns in these conditions. We recognize that the evidence connecting obesity to similar neuropsychological profiles may seem preliminary. We have tried to elaborate more clearly on how we reached our hypotheses in the revised version of the introduction.

‘Based on the above considerations these inconsistencies may be due to prior studies not clearly differentiating between distractor-resistant maintenance and updating in the context of working memory. This distinction may be crucial, however, as indirect evidence hints at potential specific alteration in these two sub-processes in obesity. For instance, obesity has been associated with aberrant dopamine transmission, with there being an abundance of literature linking obesity to changes in D2 receptor availability in the striatum (see e.g. Horstmann et al., 2015). However, results are not consensual, with studies reporting decreased, increased, or unchanged D2 receptor availability in obesity (Ribeiro et al., 2023; Janssen & Horstmann, 2022; see Darcey et al. (2023) for a potential explanation). Additionally, there are reports of differences in dopamine transporter (DAT) availability in both obese humans (Chen et al., 2008; but also see Pak et al., 2023) and rodents (Narayanaswami et al., 2013; Jones et al., 2021; Hamamah et al., 2023). The observed changes in dopamine are often interpreted as being due to chronic dopaminergic overstimulation resulting from overeating (Volkow & Wise, 2005; Volkow et al., 2008) and altered reward sensitivity as a consequence thereof (Blum et al., 1996). Considering that working memory gating is highly dependent on dopamine signaling, such changes could theoretically alter the balance between maintenance and updating processes in obesity. Next to this, obesity has frequently been associated with functional and structural changes in WM gating-related brain areas, implying another pathway through which working memory gating might get affected. At the level of the prefrontal cortex (PFC), studies have reported reduced gray matter volume and compromised white matter microstructure in individuals with obesity (Debette et al., 2014; Kullmann et al., 2016; Morys et al., 2024; Lv et al., 2024), and functional changes become evident with frequent reports of decreased activity in the dorsolateral PFC during tasks requiring cognitive control (e.g., Morys et al., 2018; Xu et al., 2017). Notably, Han et al. (2022) observed significantly lower spontaneous dlPFC activity during rest, potentially indicating reduced baseline dlPFC activity in obesity. On the level of the striatum, gray matter volume seems to correlate positively with measures of obesity (Horstmann et al., 2011), and individuals with obesity show greater activation of the dorsal striatum in response to high-calorie food stimuli compared to normal-weight individuals, indicating a stronger dopamine-dependent reward response to food cues (Stice et al., 2008; Small et al., 2003). Additionally, changes in connectivity between and within the striatum and PFC in obesity, both structurally (Li et al., 2023) and functionally (Verdejo-Román et al., 2017a; [130]; Contreras-Rodríguez et al., 2017) have been reported. Although these studies mostly investigate brain function in relation to food and reward processing, changes in these areas may also impair the ability to adequately engage in working memory gating processes, as activity in affective (reward) and cognitive fronto-striatal loops immensely overlap (Janssen et al., 2019). On the behavioral level, individuals with obesity consistently demonstrate impairments in food-specific (Janssen et al., 2017) but also non-food specific goal-directed behavioral control (Janssen et al., 2020) and reinforcement learning (Weydmann et al., 2023). It seems that difficulties with integrating negative feedback may be central to these alterations (Mathar et al., 2017; Kastner et al., 2017), which could explain a potential insensitivity to the negative consequences associated with (over) eating. Crucially, in humans, a substantial contribution to (reward) learning is mediated by working memory processes (Moustafa et al., 2008; Collins & Frank, 2012, 2018; Collins et al., 2014, 2017; Westbrook et al., 2024). The observed difficulties in reward learning in obesity may hence partly be rooted in a failure to update working memory with new reward information, suggesting cognitive issues that extend beyond mere difficulties in valuation processes. However, empirical support for this interpretation is currently lacking. A more nuanced understanding of the effects of obesity on working memory is crucial, however, as it could lead to more targeted intervention options.’

The result that Taq1A and DARPP-32 moderated the interaction between WM condition and BMI requires intricate post hoc analysis to understand the bearings to update. The authors found that Taq1A or DARPP32 genotype moderated the negative association between BMI and WM exclusively in the update condition (significant two-way interaction effect), suggesting that the BMI-WM associations in other conditions were similar across genotypes. Importantly, visual inspection of the relationship between WM and BMI (Fig 4 & 5) suggests more prevalent positive effects of the putatively advantageous Taq1A-A1 and DARPP-32-AA genotypes to the overall negative relationship between WM and BMI in updating, but not in the other conditions. Given that an overall negative relationship was statistically supported across all conditions (model 1), a plausible interpretation would be that the updating condition stands out in terms of a positive moderation by putative advantageous genotypes, rather than compound negative consequences of BMI and genotype in updating. Critically, this interpretation stands in stark contrast with the interpretation put forth by the authors suggesting a specifically negative association between BMI and WM updating.

We are grateful for the reviewers’ thorough review and insightful comments. We appreciate the attention to detail and the opportunity to improve our manuscript. We agree that further examination of the relationship between Taq1A, DARPP-32, and BMI, particularly in the update condition, is crucial for a comprehensive understanding of our results. In response to your feedback, we have conducted additional post hoc analyses, which indeed revealed the effects anticipated by the reviewer. Accordingly, we have revisited our discussion and conclusions to ensure that they accurately reflect the complexities of our findings, particularly regarding the positive moderation by putative advantageous genotypes in the update condition. Once again, we appreciate your thoughtful review and are grateful for the opportunity to strengthen the manuscript based on your feedback.

In the results section we added:

’Further post hoc examination of the effects on updating revealed that, the association between BMI and performance was significant for A1-carriers (95%CIs: -0.488 to -0.190), with 33.9% lower probability to score correctly per unit change in BMI, but non-significant for non-A1-carriers (95%CIs: -0.153 to 0.129; 1.22% lower probability). Interestingly, compared to all other conditions, in the update condition, the negative association between BMI and task performance was weakest for non-A1-carriers (estimate = -0.012, SE = 0.072, but strongest for A1-carriers estimate = -0.339, SE = 0.076; see Figure 3 and Table S6), emphasizing that genotype impacts this condition the most. To further check if this difference in slope was statistically significant across conditions, we stratified the sample into Taq1A subgroups (A1+ vs. A1-) and assessed whether BMI affected task performance differently across conditions separately for each subgroup. This analysis revealed no significant difference in the relationship between BMI and task performance across conditions among A1+ individuals (pBMI*condition = 0.219). However, within the A1- subgroup, a significant interaction effect between BMI and condition emerged (pBMI*condition = 0.049). Collectively, these findings suggest that the absence of the A1-allele is linked to improved task performance, particularly in the context of updating, where it seems to mitigate the otherwise negative effects of BMI.’

’Once more, further examination of the observed DARPP-32, BMI, and condition interaction showed that, in the update condition, the negative association between BMI and task performance was weakest and nonsignificant for A/A (estimate = -0.044, SE = 0.066; 95%CIs: -0.174 to 0.086), but strongest and significant for G-carrying individuals (estimate = -0.324, SE = 0.079; 95%CIs: -0.478 to – 0.170). See Table S7 and Figure 5. Splitting the sample in to DARPP subgroups (A/A vs. G-carrier) revealed that in both subgroups, there was significant interaction effect of BMI and condition on task performance (pA/A = 0.034, pG-carrier = 0.003). In the case of DARPP, it hence appears that carrying the disadvantageous G-allele could exacerbate the negative effects of BMI, while the more advantageous allele (A/A) might mitigate them - once again particularly in the context of updating.’

Following from this, we added the following text snippets to the discussion:

’Noteworthy, our data revealed that differences in updating appeared to be driven by the non-risk allele groups. Despite increasing BMI, performance remained stable.’

’However, as BMI increases, the possession of a greater D2 receptor density seems to become advantageous, as evidenced by the lack of a negative correlation between BMI and updating performance in non-A carriers. We speculate that this phenomenon could potentially be attributed to the compensating effects of this genotype. While individuals with fewer D2 receptors (A1+) may have quicker saturation of receptors regardless of dopamine levels, in those with more D2 receptors (A1-) saturation may be slower. This could contribute to a more finely tuned balance between ’go’ and ’no-go’ signaling, despite potential alterations in dopamine tone in obesity (Horstmann et al., 2015; but also see Darcey et al., 2023 or Janssen & Horstmann, 2022). Clearly, the current data cannot provide empirical evidence for these speculations, and further discrete research is needed to establish firm conclusions.

Regarding DARPP, we found that carrying the G-allele significantly exacerbated the negative effects of BMI, while the more advantageous allele (A/A) mitigated them, once again particularly in the context of updating.’

’Collectively, our observations hint at the potential of advantageous genotypes to moderate the adverse impacts of high BMI on cognitive functions.’

In conclusion, in its current form the title of the present work is ambivalent in terms of (1) the use of the term ’impaired’ in the context of cognitively normal individuals, (2) a BMI group difference specifically in the updating condition, and (3) the dopaminergic mechanisms based on observational data

Given the results of the additional post hoc analyses, we agree with the reviewer and have refined the title of our work to be less misleading. The title now reads:

’Working Memory Gating in Obesity is Moderated by Striatal Dopaminergic Gene Variants’

**Reviewer #1 (Recommendations for the Authors):**
Beyond the issues raised in the public review, I recommend the authors adjust the use of pathologizing terminology in the context of a clinically healthy population. In particular, terms like ’dopaminergic abnormalities’ and ’working memory deficits/impairment’ seem pathologizing in a healthy, non-morbidly obese cohort. To that end, despite a negative continuous association between BMI and WM, there are high and low-performing individuals in all BMI segments, and group differences (high vs low BMI; not reported) do not seem as dramatic as between healthy controls and say Parkinson's disease patients. Furthermore, owing to the observational design of the present study the authors should pay attention to the use of terms suggesting causal relationships, such as ’influence’ in the context of statistical associations. Also, sentences like ’Our study is the first to show such selective effects’ seem problematic not only in terms of claims of primacy, but also in terms of the selectivity of the effects (associations). See the public review for an alternative interpretation of selectivity to updating conditions.Of minor importance are the occasional spelling errors, that should be carefully checked by the authors. Also, I would like the authors to double-check the model configurations reported in the main text and the supplementary material. According to the supplement model 1 contains task condition by subject as a random effect (random slope model), whereas the main text states that this model configuration didn't converge and therefore only subject-specific intercepts are included. Hence, there seems to be discordance between the model descriptions in the main text and supplement. To that end, it would seem appropriate to briefly motivate the use of LME and the random effect for subject (within-subject correlation between conditions). Also, the origin of the odds ratios (OR) reported in the results section is not explicitly defined in the methods or results.

We appreciate the reviewer's thoughtful recommendations and have taken several steps to address the concerns raised:

(1) We have revised our manuscript to ensure that the language is less pathologizing and avoids suggesting causal relationships where only associations are indicated.

For example:

In the abstract, we replaced ’abnormalities’ with ’alterations’:

’Dopaminergic alterations have emerged as a potential mediator. However, current models suggest these alterations should only shift the balance in working memory tasks, not produce overall deficits’

In the introduction we replaced ’impairments’ with ’alterations’:

’This distinction may be crucial, however, as indirect evidence hints at potential specific alteration in these two sub-processes in obesity.

Generally, we took care to replace terms like 'dopaminergic abnormalities' and 'working memory deficits/impairments' with more neutral descriptors suitable for a clinically healthy population in the whole manuscript.

(2) We have modified primacy statements to be more nuanced. In the discussion, for example, we now say “This finding is compelling as it demonstrates a rarely observed selective effect.’ Instead of ’This finding is compelling as we are the first to show such selective effects.’

(3) We have conducted an additional thorough review of our manuscript to correct any spelling errors.

(4) Upon reevaluation, we corrected the inconsistencies with respect to the random structure of model 1. We therefore have revised the supplementary material to now accurately reflect that the model did not converge when including condition as a random factor, and thus, only subject-specific intercepts are included.

(5) We have expanded our methods section to better explain the use of linear mixed effects models (LMEs) and the inclusion of random effects for subjects to account for within-subject correlation between conditions. We added the following text:

’Given the within-subject design of our study, we used generalized linear mixed models (GLM) […]’ and

’The random structure of the model was thus reduced to include the factor ‘subject’ only, thereby accounting for the repeated measures taken from each subject.’

(6) We have clearly defined the derivation of the odds ratios reported in our results in the methods section of our manuscript. We added the following text to the methods section:

’Reported odds ratios (OR) are retrieved from exponentiating the log-odds coefficients called with the summary() function.’

**Reviewer #2 (Public Review):**
The majority of participants seem to fall within the normal BMI range, whereas the interaction between BMI and genetic variations or amino acid ratio particularly surfaces at higher BMI. As genetic variations are usually associated with small effect sizes, the effective sample size, although large for a behavioral analysis only, might have been too small to detect meaningful effects of risk alleles of COMT and C957T.

We thank the reviewer for the valuable feedback. We concur that the effective sample size may have posed a limitation in detecting meaningful effects of COMT and C957T, particularly given the skewness of our data towards participants within the normal BMI range. In response to the reviewer’s comments, we have refined the relevant paragraph in the limitations section of our manuscript, emphasizing the importance of recruiting a more balanced sample, including individuals with higher BMI, in future studies.

’Furthermore, an additional limitation is that our data is slightly skewed towards participants within the normal BMI range. The effective sample size to detect meaningful genotype effects (e.g. for COMT or C957T) might thus have been too small, particularly at higher BMI levels. Future studies may address this limitation by recruiting a more balanced sample, including more individuals with higher BMI.’

The relationships between genetic variations, BMI, and specific disturbances in dopamine signaling are complex, as compensating mechanisms might be at play to mitigate any detrimental effects. The results would therefore benefit from more direct measures or manipulations of dopaminergic processes.

We thank the reviewer for this valuable input. We acknowledge the potential benefits of employing a more direct measure, or ideally, a dopaminergic manipulation, to establish a clearer causal link between dopamine processes and working memory gating in the context of obesity. In response to the reviewers' constructive feedback, we have addressed this limitation in the discussion section of our manuscript, emphasizing the need for further research in this area:

‘Additionally, the correlational nature of our findings highlights the need for more direct experimental manipulations of dopaminergic processes in obesity. Previous studies have established a causal link between dopamine and WM gating through drug manipulations (40; 41) . Applying a similar approach to an obese sample could help establish a clearer causal link between dopamine activity and WM gating in the context of obesity.’

The introduction could benefit from a more elaborate description of the predicted effects: into which direction (better or worse updating) would the authors predict each effect to go and why? This is clearly explained for COMT, but not for e.g. DARPP-32.

We thank the reviewer for their valuable feedback. We appreciate the suggestion to provide a more detailed description of the predicted effects for each genetic marker in the introduction. We would like to note, however, that the analyses involving markers such as DARPP-32 were inherently exploratory in nature. Consequently, we intentionally refrained from formulating directed hypotheses, as our primary aim was to observe and report any emergent patterns.

**Reviewer #2 (Recommendations for the Authors):**
To what extent are the polymorphisms or amino acid ratios associated with BMI? For example, when including C957T polymorphism in the analysis, the detrimental effect of BMI on working memory is no longer statistically significant. Could this be due to a relatively strong relationship between C957T polymorphism and BMI? Could the authors provide figures showing how BMI relates to the genetic polymorphisms and amino acid ratio?

We appreciate the reviewer's insightful comment and have thoroughly investigated the potential relationship between the polymorphism and BMI. Our analysis did not reveal any direct association between C957T and BMI. We have included this analysis in our manuscript. The reviewer’s comment strengthened the comprehensiveness of our study.

’Because the main effect of BMI dissipated when including C957T in the model, we ran an additional exploratory analysis to check whether this polymorphism directly related to BMI. Linear regression, predicting BMI by genotype, showed no association between the two (p = 0.2432), indicating that BMI effect is probably not masked by the presence of the C957T polymorphism. See Table S8.’